# Potential of Erythrocyte Membrane Lipid Profile as a Novel Inflammatory Biomarker to Distinguish Metabolically Healthy Obesity in Children

**DOI:** 10.3390/jpm11050337

**Published:** 2021-04-23

**Authors:** Iker Jauregibeitia, Kevin Portune, Itxaso Rica, Itziar Tueros, Olaia Velasco, Gema Grau, Luis Castaño, Federica Di Nolfo, Carla Ferreri, Sara Arranz

**Affiliations:** 1AZTI, Food Research, Basque Research and Technology Alliance (BRTA), Parque Tecnológico de Bizkaia, Astondo Bidea, Edificio 609, 48160 Derio, Spain; ijauregibeitia@azti.es (I.J.); kportune@azti.es (K.P.); itueros@azti.es (I.T.); 2Biocruces Bizkaia Health Research Institute, Cruces University Hospital, CIBERDEM/CIBERER, UPV/EHU, Endo-ERN, 48903 Barakaldo, Spain; itxaso.ricaechevarria@osakidetza.eus (I.R.); olaia.velascovielba@osakidetza.eus (O.V.); mariagema.graubolado@osakidetza.eus (G.G.); luisantonio.castanogonzalez@osakidetza.eus (L.C.); 3Lipidomic Laboratory, Lipinutragen srl, Via di Corticella 181/4, 40128 Bologna, Italy; federica.dinolfo@lipinutragen.it; 4Consiglio Nazionale delle Ricerche, ISOF, Via Piero Gobetti 101, 40129 Bologna, Italy

**Keywords:** children, inflammation, lipidomics, mature erythrocyte membrane, metabolically healthy obesity

## Abstract

Metabolically healthy obesity (MHO) has been described as BMI ≥ 30 kg/m^2^, without metabolic disorders traditionally associated with obesity. Beyond this definition, a standardized criterion, for adults and children, has not been established yet to explain the absence of those metabolic disorders. In this context, biomarkers of inflammation have been proposed as suitable candidates to describe MHO. The use of mature red blood cell fatty acid (RBC FA) profile is here proposed since its membrane lipidome includes biomarkers of pro- and anti-inflammatory conditions with a strict relationship with metabolic and nutritional status. An observational study was carried out in 194 children (76 children with obesity and 118 children with normal weight) between 6 and 16 years old. RBC FAs were analyzed by gas chromatography-flame ionization detector (GC-FID). An unsupervised hierarchical clustering method was conducted on children with obesity, based on the RBC FA profile, to isolate the MHO cluster. The MHO cluster showed FA levels similar to children with normal weight, characterized by lower values of arachidonic acid, (total ω-6 FA, ω6/ω3 FA ratios and higher values for EPA, DHA, and total ω-3 FA) (for all of them *p* ≤ 0.01) compared to the rest of the children with obesity (obese cluster). The MHO cluster also presented lipid indexes for higher desaturase enzymatic activity and lower SFA/MUFA ratio compared to the obese cluster. These differences are relevant for the follow-up of patients, also in view of personalized protocols providing tailored nutritional recommendations for the essential fatty acid intakes.

## 1. Introduction

Obesity is a growing health problem affecting children and adolescents. According to the WHO states, with more than 340 million children and adolescents, around one in three from 5 to 19 years old were in condition of overweight or obesity in 2016, and 38 million children under the age of 5 were overweight or obese in 2019 [1]. Furthermore, children with obesity tend to be obese later in life, increasing their risk for morbidity and mortality [2].

Within this large patient cohort, there is an emerging evidence that not all of them display common obesity-associated metabolic disorders, such as insulin resistance, glucose intolerance, arterial hypertension, or dyslipidemia, suggesting that a paradox of metabolically healthy obesity (MHO) can exist, independent of fat accumulation [3]. Indeed, although most people with obesity are metabolically unhealthy, a percentage varying from 10% to 30% of those with obesity [4,5,6,7,8] are considered MHO and do not present metabolic abnormalities [9].

Even though the existence of MHO has been known for decades [4], there is still a lack of a proper definition of metabolically healthy obese individuals. Some studies describe MHO as BMI ≥ 30 kg/m^2^, without metabolic disorders (e.g., type 2 diabetes or dyslipidemia) [3,4,9] while others take into account their body fat percentage and insulin resistance [10]. The MHO phenotype has been associated with a good metabolic profile characterized by high levels of insulin sensitivity, low prevalence of hypertension, and a favorable fasting glucose, lipid, and inflammatory profile [11]. However, no consensus has been reached to define a standardized criterion to categorize MHO in adults. Recent investigations showed the importance of including inflammatory parameters such as circulating blood proinflammatory cytokines, adipokines, and acute-phase response proteins as possible biomarkers to define MHO [12,13] that have been studied as possible underlying factors of inflammation [14].

Pediatric MHO was recently defined [15] as those children characterized by the absence of traditional cardiometabolic risk factors. However, recent studies also focused on considering other parameters such as hepatic steatosis, inflammatory biomarkers, as well as the degree of visceral fat accumulation [16], as is considered in adult populations. Obesity in children has been also associated with circulating inflammatory mediators [17], where essential fats, such as ω-6 and ω-3 FAs, have a direct impact on the onset and control of metabolic pathways of inflammation [18].

Currently, there are only limited longitudinal studies on pediatric population evaluating MHO according to inflammatory biomarkers, and this is a limitation both in the clinical evaluation and in the personalization of the therapeutical approach, including nutritional intervention strategies for children with obesity, according to their metabolic health status.

In this regard, the use of mature erythrocyte as a representative site for all other body tissues, in particular determining the fatty acid-based membrane lipidomic profile, is an established protocol for membrane-based molecular diagnostics [19,20,21] and can help with monitoring the ω-6 and ω-3 FA contents in phospholipids that are directly linked with the inflammation mediators. Arachidonic acid, which is the main ω-6 FA in the RBC membrane, is released from membrane phospholipids, and exerts its activity in defense mechanisms, but the presence of a continuous inflammatory stimulus can induce an excess of inflammatory mediators thus triggering inflammation as a chronic status. The unbalance created by essential FA deficiency in diet, mainly ω-3 FA, is known to be connected to many symptoms and tissue malfunctions [22]. On this basis, it is clear that the membrane RBC fatty acid profile can ultimately define the status of “silent inflammation” in obesity patients. It is worth adding the role of a key fatty acid in ω-6 pathway, namely, the AA precursor, dihomo-gamma linolenic acid (DGLA), that is well known for its anti-inflammatory effects through the transformation into series 1 prostaglandins [23].

Moreover, a balance between ω-6 and ω-3 FA has been described as a good marker for inflammation status in a subject [24,25].

In view of the intense research on molecular medicine and the set-up of nutritional intervention studies to better understand the impact of personalized diet on lipid metabolism in children, we were interested in understanding the role of mature RBC lipid profile to individuate metabolic differences in patients with obesity, extending previously published works on this subject [25,26,27]. In this previous works, children with obesity showed a pro-inflammatory RBC FA profile, specially, due to increased values of AA and an unbalanced saturated fatty acid (SFA)/monounsaturated fatty acid (MUFA) ratio towards the SFA pathway.

In this study, we used the mature RBC FA profile as a comprehensive biomarker to individuate the pro- and anti-inflammatory conditions and distinguish metabolically healthy obesity in children in relation with their nutritional status.

## 2. Materials and Methods

### 2.1. Subjects and Study Design

An observational study was conducted on 194 children (76 children with obesity and 118 children with normal weight) between 6 and 16 years old, recruited from the pediatric endocrinology unit at the Hospital Universitario Cruces (Barakaldo, Spain). Children were classified according to body mass index (BMI), using age- and sex-specific pediatric z-scores from Orbegozo table [28]. BMI was taken as a reference to define the different categories, defining normal weight when the standard deviation (SD) of BMI was −1 < SD ≤ +1 and obesity when SD > +2. Groups were homogeneously distributed by age.

Subjects were excluded based on the following criteria: they presented any kind of acute or chronic diseases, were taking medications, had any presence of metabolic syndrome symptoms, or if their obesity condition was associated to any type of pathology. An anthropometric examination was performed by an endocrinologist.

The study protocol was approved by the Euskadi Clinical Research Ethics Committee (permission number PI2016181) and accomplished according to the Helsinki Declaration in 1975, revised in 2013. Subjects under study were included after acceptance (by the parents of the individuals) to participate in the study and signing of informed consent. All the informed consent documents were signed by their parents, and in the case of children between 12 and 16 years of age, the informed consent was also signed by themselves according to the Euskadi Ethical Committee and sample biobank laws (Organic Law 3/2018, of December 5, on Protection of Personal Data and guarantee of digital rights; Law 14/2007 on Biomedical Research and RD 1716/2011 of Biobanks).

### 2.2. Anthropometric Measures

Body weight (kg) and height (cm) were measured by standardized methods [29]. Body mass index (BMI) was calculated as weight (kg) divided by the square of the height (m^2^). Anthropometric parameters, as well as blood sampling, were all conducted by pediatricians during the participant’s visit to the Hospital Universitario Cruces.

### 2.3. Nutrient Intakes

During the first visit, a pediatrician interviewed the participants and collected personal data, including family medical history and information on the history of medication usage. Evaluation of dietary habits and estimations of food consumption, including dietary diversity and variety, were measured using different validated questionnaires already described in previous publications [30,31,32,33,34].

### 2.4. Red Blood Cell (RBC) Membrane Fatty Acid Analysis

The fatty acid composition of mature RBC membrane phospholipids was obtained from blood samples (approximately 2 mL) collected in vacutainer tubes containing ethylenediaminetetraacetic acid (EDTA). Samples were shipped to the company Lipinutragen (Bologna, Italy) and upon arrival underwent quality control for the absence of hemolysis. During the blood analysis, an automated protocol consisted of selection of mature RBCs followed by lipid extraction and lipid transesterification to fatty acid methyl esters (FAMEs) was used [21,35,36,37,38,39], as reported previously in our previous studies [25].

### 2.5. Red Blood Cell Membrane Fatty Acid Profile

A pool of 12 FAs were selected as a representative profile of the dominant glycerophospholipids present in the RBC membrane, as well as three FA families (SFA, MUFA, and PUFA): for SFAs, palmitic acid (C16:0) and stearic acid (C18:0); for MUFAs, palmitoleic acid (C16:1;9c), oleic acid (C18:1; 9c), and cis-vaccenic acid (C18:1; 11c); for ω-3 PUFAs, eicosapentaenoic acid (EPA) (C20:5) and docosahexaenoic acid (DHA) (C22:6); for ω-6 PUFAs, linoleic acid (LA) (C18:2), dihomo-gamma-linolenic acid (DGLA) (C20:3), and arachidonic acid (AA) (C20:4); and for geometrical trans fatty acids (TFA), elaidic acid (C18:1 9t) and mono-trans arachidonic acid isomers (monotrans-C20:4; ω-6 recognized by standard references as previously described by Ferreri et al.) [40]. Considering these fatty acids, different indexes previously reported in the literature [37] were calculated: (%SFA/%MUFA) index related with membrane rigidity; Omega-3 index (DHA + EPA); Inflammatory risk index (% ω-6)/(% ω-3); PUFA balance ((%EPA + %DHA)/total PUFA × 100); Free radical stress index (sum of trans-18:1 + Σ monotrans 20:4 isomers); Unsaturation Index (UI) ((%MUFA) + (%LA/2) + (%DGLA/3) + (%AA/4) + (% EPA/5) + (%DHA/6)); Peroxidation Index (PI) ((%MUFA/0.025) + (%LA) + (%DGLA/2) + (%AA/4) + (% EPA/6) + (%DHA/8)); De Novo Lipogenesis index (DNL) ((%Palmitic acid)/(%LA)) [41].

Additionally, the enzymatic indexes of elongase and desaturase enzymes, the two classes of enzymes of the MUFA and PUFA biosynthetic pathways, were inferred by calculating the product/precursor ratio of the FAs involved in these reactions.

### 2.6. Statistical Analysis

In order to classify individuals based on metabolic similarities in their fatty acid profile, an unsupervised hierarchical clustering method was conducted on children with obesity, based on the RBC FA profile, using SPSS v.25 (IBM, Chicago, IL, USA). The idea of cluster analysis is to measure the distance between each pair of objects (participants) in terms of the variables suggested in the study (PLM FA levels) and then to group subjects, which are close together. More specifically, based on the distance matrix, the clustering algorithm identifies the closest observations (i.e., subjects with similar RBC FA profile levels) and iteratively merged them within the same cluster until all clusters were merged together [42]. The result is a hierarchical classification tree [43] (Figure 1).

The clustering was performed based on the method of Ward (1963), which was found to be most suitable as it creates a small number of clusters with relatively more participants. Additionally, the Ward method has proved to outperform other hierarchical methods (Punj and Stewart, 1983; Harrigan, 1985) in producing homogeneous and interpretable clusters.

Once the MHO cluster was isolated from the rest of the children with obesity (the rest of the obese clusters that were not the MHO cluster were merged to form the obese cluster), statistical analyses was made between the MHO cluster, the obese cluster, and the children with normal weight groups.

Differences between the MHO cluster, obese cluster, and children with normal weight for nutrient intake, food group intake, and KIDMED test were determined by conducting a Kruskal–Wallis test for the data that were not normally distributed. Normal data distribution was assessed by Shapiro–Wilk’s test or/and Kolmogorov–Smirnov test. Subsequently, Dunn’s (1964) test was performed for post hoc comparisons. A Bonferroni correction for multiple comparisons was made, to correct for the increased risk of type I error. For normally distributed variables, a one-way ANOVA with Tukey post hoc analysis was conducted.

An Analysis of Covariance Test (ANCOVA) was run to determine the differences between RBC membrane fatty acids from the obese cluster, MHO cluster and children with normal weight, after controlling for variables selected as potential confounders, such as age, gender, and dietary macro- and micronutrient intake. Post hoc analysis was performed with a Bonferroni adjustment for multiple comparisons. First, a principal component analysis (PCA) was run on 15 dietary nutrient intake variables (individual FAs, families (SFA, MUFA, and PUFA), total lipids (%E), and carbohydrates and proteins), obtained with the DIAL software after transforming the information about food items from FFQ questionnaires into micro- and macronutrient values, in order to reduce and simplify the dimensions of these variables and use the generated factors as diet covariates [37]. The Kaiser–Meyer–Olkin (KMO) and Bartlett’s test of sphericity were used to verify the sampling adequacy for the analysis. PCA revealed four components that had eigenvalues greater than one and which explained 83.74% of the total variance. These components were included in the ANCOVA analysis as diet covariates. The level of significance was set at *p* < 0.05. All statistical analyses were performed using SPSS (IBM Corp. V 24.0, New York, NY, USA).

## 3. Results

### 3.1. Clustering

Although selection of participants was done according to z-score tables to identify 2 phenotypes (normal weight and obesity children) as we have explained in Materials and Methods section, authors performed a hierarchical clustering using the squared Euclidean distance and Ward’s method to classify each subject based on the 12 FA measured in the RBC profile. In the total sample of obesity children, five clusters were isolated (Figure 1). Then, the RBC-FA profile of these five clusters were compared with the RBC-FA profile of the normal weigh children’s group. After that, one of them, presented a similar RBC-FA profile as the normal weight group, which was named “the metabolically healthy obese group (MHO).” As the other four clusters did not show significant differences in the RBC FA profile compared to the normal weight group, they were merged into one group as the obese cluster. A hierarchical clustering was performed.

### 3.2. Descriptive Characteristics of the Clusters

A total of 194 children between 6 and 16 years old took part in the study sample. The MHO cluster and the obese cluster together with normal weight group were included in a comparative analysis to describe their characteristics (Table 1). A matched gender distribution was found for the obese cluster and MHO but not for the normal weight group. Three groups presented similar age without statistically significant differences. No variation was observed for BMI between the obese cluster and the MHO cluster.

### 3.3. Red Blood Cell Membrane Fatty Acids Profile

In order to compare RBC FA profiles between groups, a one-way ANCOVA was conducted using age, sex, and dietary intake as covariates to adjust the error made by those confounding factors (Table 2). No statistically significant difference was observed between the MHO cluster compared with the control group, apart from the 20:4 trans FA that showed higher levels for the MHO cluster (*p* ≤ 0.001).

The obese cluster presented significant differences when it was compared with both MHO cluster and the control group. The obese cluster presented higher values for total SFA, AA, total ω-6, ω6/ω3, SFA/MUFA, and D9D 18:0 and lower values for oleic acid, total MUFA, EPA, DHA, total ω-3, D6D+ELO, and PUFA balance. The obese cluster also had higher values of DGLA and stearic acid compared to the normoweight group (*p* ≤ 0.001 for both), but no differences with the MHO cluster were observed (*p* = 0.08 for both). The obese cluster presented lower levels of 20:4 trans FA compared to the MHO cluster (*p* ≤ 0.001), but no differences with the normoweight group were observed.

### 3.4. Dietary Intake

Table 3 shows the differences in macronutrients and individual fatty acid daily intake expressed as % of Kcal among the three groups. No statistically significant differences were observed for any of measured macro- and micronutrients intake except for total PUFA, for which obese cluster showed lower intake compared to the normoweight group (*p* = 0.03).

### 3.5. Food Groups

Table 4 shows dietary intake according to food categories calculated via food frequency questionnaires. The MHO cluster showed a significantly higher consumption of fruits than the obese cluster (*p* = 0.01) and the normoweight group (*p* = 0.02). The obese cluster presented a lower intake of cereals compared with the normoweight group (*p* = 0.04) and a lower score from the KIDMED test (*p* = 0.02). No other differences were observed regarding food groups intake.

## 4. Discussion

To our knowledge, this is the first time that an RBC FA membrane profile has been used as a biomarker to differentiate MHO in a cohort of obese patients, and especially in children, that present metabolic imbalances. Inflammation seems to play a key role in distinguishing metabolically healthy from metabolically unhealthy individuals with obesity [3], so here, the analysis of an RBC FA profile as a comprehensive biomarker of the pro- and anti-inflammatory status is assayed, which can be applied as a potential tool to distinguish MHO.

Previous studies in MHO adults and children have been focused on biochemical parameters such as insulin resistance, blood pressure, serum lipids, and glucose [4,5,6,7,8], but it has been increasingly suggested to include circulatory inflammatory markers in the definition of MHO [12,44].

The RBC FA profile characteristic of children with obesity has been previously reported in the literature [25], compared to children with normal weight, with a clear shift of the group with obesity towards a pro-inflammatory condition, mainly due to the higher levels of arachidonic acid, a well described precursor of inflammatory mediators [45,46], accompanied by a higher SFA/MUFA ratio.

After conducting a hierarchical clustering analysis of our patient cohort, within those individuals with obesity, an MHO cluster was isolated, which showed a differentiated FA profile, compared to the rest of the children with obesity, and, similar profile to children with normal weight. Our MHO cluster matched the previously published prevalence of 10–30% of the study population (14% in our study) [4,5,6,7,8].

Compared to children with obesity, the MHO cluster, showed lower levels of total ω-6 FA, mainly due to significant lower value of AA (*p* < 0.001), a well-known precursor of proinflammatory mediators prostaglandins, thromboxane A2, and prostacyclins [45,46]. The MHO cluster does not present a shift to an inflammatory metabolism, as with other children with obesity do, and displays values of ω-6 FAs similar to children with normal weight.

Regarding ω-3 FAs, the MHO cluster presents higher values of EPA, DHA, and total ω-3 FAs (*p* < 0.001, *p* = 0.001 and *p* < 0.001 consecutively) compared to children with obesity with similar values compared to the normal weight children. This is an important feature as, ω-3 PUFAs can improve impaired metabolism in obesity by modulating main metabolic pathways [47], such as promoting anti-inflammatory response or insulin sensitivity [48], regulating the adipocyte apoptosis [49], or modulating membrane fluidity by altering lipid rafts [50]. In fact, a balanced ω-6/ω-3 ratio is important in the prevention and management of obesity, as both metabolic pathways compete to bind the same enzymes and an unbalanced ratio towards the ω-6 PUFAs appears remarkably enhanced in obesity [24].

Regarding SFA, the MHO cluster showed lower levels of total SFA in RBC membranes (*p* = 0.03) and higher values of total MUFA (*p* = 0.01), mainly due to higher levels of oleic acid (*p* = 0.04). However, these differences were not associated to different dietary intake of SFA or MUFA, as could be considered a priori, because no differences were observed for any nutrient intake between the MHO cluster and the rest of children with obesity. The MHO cluster only showed lower intake for total PUFA intake (*p* = 0.03), which could be associated to lower levels of total PUFA in RBC for MHO cluster (*p* = 0.03). However, dietary variables that can act as confounder factors, have been used as covariates in the ANCOVA analysis, as explained in the experimental section, to observe differences in RBC profile between groups. Moreover, despite the food frequency questionnaires used in this study are validated and widely used, they have their limitations to describe accurately diet intake and this could be seen as a limitation of the study [51].

Regarding food groups intake, only a higher intake of fruits was observed (*p* = 0.01) for MHO compared to other children with obesity, which can be considered a protective factor together with the consumption of vegetables due to a higher consumption of fiber and low glycemic load [52], since it can provide an antioxidant intake that exert their protection toward the FA in the lipid pools, especially PUFA. However, this could be enough explaining the FA composition in our cohort, considering that no differences were observed for the KIDMED score.

As practically no differences were seen in terms of reported intakes, our attention turned to evaluate enzymatic activity using lipid indexes in order to explain the differentiated levels of RBC SFA and MUFA. The enzymatic activity of the Δ-9-desaturase appeared higher for the MHO compared to the rest of the children with obesity (*p* < 0.001) and indicated the higher proportions of SFAs converted to MUFAs, as reflected in higher oleic acid levels and almost statistically significant lower stearic acid levels (*p* = 0.08) in this group. A reduced activity of the enzyme in the obese cluster is correlated to factors that have been recalled several times to explain the involvement of SFA pathway in metabolic derangements, such as the absence of enzymatic cofactors, the inhibition of desaturase activity, and liver impairment [22], since the desaturase transformation prevents SFA accumulation and toxicity-triggering hepatocellular apoptosis and liver damage [53].

Previous author’s publication revealed the use of RBC FA profile as a biomarker to identify differences between children with obesity compared to overweight and normal weight children in order to design personalized and specific dietary recommendations mainly for dietary fats [25]. These results are aligned with RBC FA profile showed in this new work for obese clusters. However, this new work highlights the importance of using cell membrane biomarkers, as the RBC FA profile, for the identification of non-inflammatory molecular profile of children with obesity that can be used by clinicians to design differentiated nutritional interventions.

It is worth mentioning that, although very popular, the indirect measurement of enzyme activity, by partition between product and precursors, could be considered as a limitation of the study, whereas the direct measurement is needed to reaffirm the conclusions obtained. Longitudinal studies should be performed in order to identify if the MHO group does not develop obesity-associated comorbidities. At the same time, an increase in the sample size would be interesting to confirm these results. Biochemical parameters (glucose, triglycerides, and cholesterol) would be interesting to be measured and check if they present differences as we observed for RBC FA.

The underlined differences between MHO cluster and the rest of children with obesity are connected with the importance of a personalized approach in obesity, in particular, regarding nutritional recommendations based on the specific FA needs. Currently, general recommendations for individuals with obesity, such as low caloric diets with restriction on fats, are given [54], but not considering the quality of fats and specifically the types of FA needed. The membrane FA lipidome analysis should be included in the protocols for two reasons: (i) to better define and differentiate the molecular status of the patients and (ii) to envisage adequate nutritional strategies for each population group and to test their effects increasing the personalization of the treatments. According to our results, individuals in the obese cluster need a higher intake of ω-3 FA, to induce a better balance between ω-6 and ω-3 pathways and reduce the inflammatory precursors. This recommendation is not extended to all children with obesity, as the MHO cluster has optimal levels of both ω-3 and ω-6 FAs.

## 5. Conclusions

The capacity of the mature RCB FA profile can be used for larger population studies, to extend its validation as a biomarker to differentiate MHO in children describe the characteristics of those children with obesity that do not display typically obesity-associated metabolic imbalances. At the same time, intervention studies, with personalized nutritional strategies, can be carried out keeping the optimal balance of the RBC FA profile as a molecular target to couple with clinical observations.

## Figures and Tables

**Figure 1 jpm-11-00337-f001:**
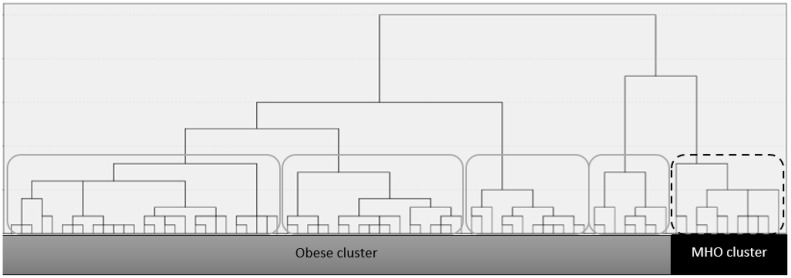
Hierarchical clustering classification tree.

**Table 1 jpm-11-00337-t001:** General characteristics of the studied population.

	Obese Clusters (G1) *n* = 65		MHO Cluster (G2) *n* = 11		Normoweight (G3) *n* = 118		Kruskal–Wallis H Test (*p*)	Post Hoc Pairwise Comparison (*p*)		
	Mean	SD	Mean	SD	Mean	SD		G1:G2	G1:G3	G2:G3
Age	11.0	0.3	10.8	0.7	10.9	0.3	0.94			
Sex (% girls)	68		72.7		46.5		0.01	1.00	0.01	0.26
BMI	28.7	0.4	29.0	1.23	18.4	0.3	<0.001	1.0	<0.001	<0.001

**Table 2 jpm-11-00337-t002:** Red blood cell (RBC) membrane fatty acid profile.

	Obese Cluster (G1) *n* = 65		MHO Cluster (G2) *n* = 11		Normoweight (G3) *n* = 118		ANCOVA	Post Hoc Pairwise Comparison (*p*-Value)		
Fatty acids (%)	Mean	SE	Mean	SE	Mean	SE	*p*	G1:G2	G1:G3	G2:G3
16:0	22.38	0.13	22.62	0.32	22.50	0.09	0.64	1.00	1.00	1.00
18:0	18.35	0.13	17.59	0.32	17.72	0.10	<0.001	0.08	<0.001	1.00
Total SFA	40.72	0.12	39.84	0.32	40.13	0.09	<0.001	0.03	<0.001	1.00
16:1/9c	0.42	0.02	0.50	0.04	0.40	0.01	0.07	0.28	0.83	0.76
18:1/9c	16.34	0.15	17.34	0.37	17.45	0.11	<0.001	0.04	<0.001	1.00
18:1/11c ^b^	1.13	0.03	1.32	0.07	1.19	0.02	0.02	-	-	-
Total MUFA	17.92	0.16	19.15	0.39	19.06	0.12	<0.001	0.01	<0.001	1.00
18:2	13.90	0.17	13.91	0.41	14.24	0.12	0.26	1.00	0.33	1.00
20:3	2.32	0.05	2.05	0.11	2.02	0.03	<0.001	0.08	<0.001	1.00
20:4	20.02	0.17	18.16	0.43	18.62	0.13	<0.001	<0.001	<0.001	0.93
Total ω6	36.23	0.20	34.12	0.49	34.94	0.15	<0.001	<0.001	<0.001	0.33
20:5	0.46	0.03	0.77	0.07	0.61	0.02	<0.001	<0.001	<0.001	0.09
22:6	4.52	0.13	5.57	0.33	5.01	0.10	0.001	0.01	0.01	0.33
Total ω3	4.92	0.15	6.33	0.36	5.63	0.11	<0.001	<0.001	<0.001	0.20
Total PUFA	41.20	0.18	40.45	0.46	40.57	0.14	0.02	0.38	0.03	1.00
18:1t	0.09	0.01	0.06	0.02	0.08	0.01	0.52	0.79	1.00	1.00
20:4t	0.06	0.01	0.13	0.02	0.07	0.01	<0.01	<0.001	0.72	<0.001
Total TRANS	0.14	0.01	0.19	0.03	0.16	0.01	0.21	0.27	1.00	0.61
Indexes										
ω6/ω3	7.65	0.21	5.58	0.51	6.47	0.15	<0.001	<0.001	<0.001	0.29
SFA/MUFA	2.28	0.02	2.11	0.06	2.13	0.02	<0.001	0.01	<0.001	1.00
Δ6D+ELO	6.09	0.16	6.91	0.38	7.20	0.11	<0.001	0.14	<0.001	1.00
Δ5D 20:4	8.87	0.24	9.06	0.59	9.43	0.18	0.18	1.00	0.20	1.00
Δ9D 16:0	58.21	2.49	51.42	6.17	59.11	1.86	0.50	0.91	1.00	0.72
Δ9D 18:0	1.13	0.01	1.01	0.03	1.02	0.01	<0.001	<0.001	<0.001	1.00
PUFA balance	12.02	0.35	15.72	0.86	13.91	0.26	<0.001	<0.001	<0.001	0.14
Peroxidation Index	137.00	1.05	140.18	2.55	137.12	0.76	0.74	0.74	1.00	0.77
Unsaturation Index	162.17	0.76	163.00	1.88	161.33	0.56	1.0	1.00	1.00	1.00
DNL Index	1.63	0.02	1.60	0.05	1.59	0.02	1.0	1.00	0.36	1.00

Data are presented as mean ± standard error. Adjusted for age, sex, and dietary components extracted from the principal component analysis of dietary nutrient intake (individual FAs, families (SFA, MUFA, and PUFA), total lipids (%E), carbohydrates, proteins, and calories). ^b^ Levene’s test of homogeneity of variance was not met.

**Table 3 jpm-11-00337-t003:** Dietary daily intake expressed as % of energy (%E).

	Obese Clusters (G1) *n* = 65		MHO Cluster (G2) *n* = 11		Normoweight (G3) *n* = 118		Kruskal–Wallis H Test (*p*)	Post Hoc Pairwise Comparison (*p*)		
Macronutrients										
	Mean	SD	Mean	SD	Mean	SD		G1:G2	G1:G3	G2:G3
Calories (Kcal/day)	2002.34	583.09	2320.70	371.52	2479.31	1811.90	0.09			
Proteins (%E)	16.54	2.16	16.34	1.89	16.37	2.68	0.96			
Carbohydrates (%E)	46.96	5.25	45.73	5.84	42.64	7.50	0.64 *			
Simple sugars (%E)	21.74	5.25	21.72	2.28	20.38	5.84	0.90 *			
Lipids (%E)	33.34	6.25	35.05	6.83	37.28	7.33	0.64			
Individual FA (% E)										
C14:0	0.95	0.43	1.17	0.73	0.91	0.39	0.31 *			
C16:0	6.07	1.21	6.58	1.74	5.93	1.18	0.17			
C18:0	2.32	0.56	2.49	0.83	2.33	0.58	0.62 *			
Total SFA	10.61	2.65	11.84	4.32	10.98	2.45	0.09			
C16:1	0.52	0.14	0.56	0.18	0.52	0.12	0.65			
C18:1	13.91	3.63	14.63	2.95	16.27	4.17	0.55 *			
Total MUFA	14.86	3.72	15.66	3.09	17.22	4.22	0.50			
C18:2	4.19	1.78	3.63	1.30	5.00	2.44	0.33			
C20:4	0.53	0.12	0.60	0.14	0.65	0.27	0.82 *			
Total ω-6	4.23	1.78	3.68	1.33	5.07	2.46	0.37			
C18:3	0.04	0.01	0.04	0.01	0.06	0.03	0.25			
C20:5 (EPA)	0.08	0.06	0.07	0.05	0.07	0.05	0.97			
C22:5 (DPA)	0.02	0.01	0.02	0.01	0.02	0.01	0.68			
22:6 (DHA)	0.15	0.10	0.14	0.08	0.15	0.09	0.96			
Total ω-3	0.78	0.22	0.82	0.24	0.90	0.33	0.46			
Total PUFA	5.14	1.84	4.67	1.48	6.11	2.67	0.01	1.00	0.03	0.12
ω6/ω3	5.70	2.34	4.57	1.19	6.00	3.20	0.05			

Data are presented as mean ± standard deviation. Not normally distributed variables. Pairwise comparison conducted with a Bonferroni adjustment. * ANOVA was conducted instead of Kruskal–Wallis. Post hoc pairwise comparisons are only shown for cases with a significant difference between FA using ANOVAs or Kruskal–Wallis tests.

**Table 4 jpm-11-00337-t004:** Food groups intake.

	Obese Clusters (G1) *n* = 65		MHO Cluster (G2) *n* = 11		Normoweight (G3) *n* = 118		Kruskal–Wallis H Test (*p*)	Post Hoc Pairwise Comparison (*p* *)		
Food Groups (g/day)	Mean	SD	Mean	SD	Mean	SD		G1:G2	G1:G3	G2:G3
Fruits	434.7	39.5	611.3	56.4	445.9	23.8	0.01	0.01	0.77	0.02
Vegetables	166.4	16.3	180.0	27.6	192.1	13.4	0.3			
Cereals	142.9	7.0	173.8	20.8	172.9	8.5	0.04	0.45	0.04	1.0
Legumes	79.0	3.7	75.0	8.3	81.2	3.6	0.86			
Olive oil	19.5	1.6	27.3	3.5	21.1	1.1	0.06			
Dairy products	340.0	28.7	271.9	37.4	360.3	15.8	0.12			
Eggs	20.9	1.4	28.5	4.1	24.4	1.9	0.28			
Red meat	30.2	2.5	36.7	6.8	27.7	1.8	0.34			
White meat	40.3	2.3	47.4	2.6	41.2	2.6	0.25			
Dried fruits and nuts	3.8	0.9	4.7	2.6	5.3	0.7	0.24			
Lean fish	29.0	2.5	41.8	7.8	31.6	1.8	0.16			
Oily fish and shellfish	28.4	2.9	32.5	6.6	26.1	2.1	0.27			
Sugary drinks	46.0	10.3	43.8	16.2	43.6	12.0	0.65			
Juices	123.8	15.0	148.5	28.4	134.9	15.2	0.68			
KIDMED	7.11	2.23	7.60	1.90	7.95	1.87	0.02	1.0	0.02	1.0

Data are presented as mean ± standard deviation. * Pairwise comparison conducted with a Bonferroni adjustment.

## Data Availability

The data presented in this study are available on request from the corresponding author (S.A.). Raw data were generated at AZTI, Biocruces Bizkaia Health Research Institute and Lipidomic Laboratory maintaining samples anonymized.

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
