# Peer review of "Potential of Erythrocyte Membrane Lipid Profile as a Novel Inflammatory Biomarker to Distinguish Metabolically Healthy Obesity in Children"

_jpm, 2021, doi:10.3390/jpm11050337_

Round 1

Reviewer 1 Report

Good introduction to the subject of the study.

- page 3: line 103 – could you please explain what does ‘z-scores from Orbegozo table’ specifically mean?

- page 5: line 237 – did you mean to say ‘Five clusters were isolated’?

- page 6: Table 2 - first row – I would suggest moving the columns a little so that the ‘Anova’ sign in the first row could be in one line, if it is possible

- page 7: I would suggest moving the first row of Table 2 to the next page as it would allow for better viewing and understanding of the data

- page 7: line 274 – ‘No statistically differences were observed’ – this wording does not seem to be correct

- page 8: line 284 & page 9: Table 3 – You spelled ‘KIDMED’ with all capital letters throughout the whole article but here you spelled it ‘Kidmed’. ‘KIDMED’ also appears in the articles you cite in your references. I would suggest being consistent with the capitalization.

Interesting choice of subject with a novel perspective.

Clear layout and description of methods and research.

Good use of English language.

Reviewer 2 Report

Jauregibeitia and colleagues investigate erythrocyte membrane lipid profile in metabolically healthy obesity (MHO).

MHO still lacks a shared definition so the author's aim would be to highlight some characteristic metabolic pattern.

I found the topic of great relevance both because the investigation is about childhood/adolescence obesity, both because the erythrocyte membrane fatty acid profile has been investigated.

The manuscript is overall well-structured but below some concerns are reported:

  1. Introduction, lines 92-93: The authors cite this work as a continuation of previous researches. I think should be useful for the readers to briefly know what has been discovered there.
  2. Introduction: Among the aims, the use of the nutrients intake assessment should be introduced.
  3. Materials and methods: Since most of the methods are exactly the same as the authors' previous work (copy and paste), some parts should be either summarized referring to their previous work or changed.
  4. Clustering: It is clear the technical part of the analysis, but I don't find clear the definition of MHO cluster based on the comparison between "RBC fatty profile similar to children with normal weight". Could please the author give more information about these criteria? Did the authors investigate other biochemical parameters?
  5. Discussion, line 332: (p=0.03) shouldn't be (p=0.01?)
  6. Discussion: In the discussion, some comparisons with the authors' previous studies could be useful to clarify how this work extends the previous study.
  7. Discussion: Some strengths and limitations should be added
  8. In general, a clearer description of what the authors consider to be a biomarker for MHO differentiation should be provided.
  9. Table 3 could be moved to supplementary material

Round 2

Reviewer 2 Report

Thanks to the authors for providing clear answers. I have no further comments.